# Clinical Outcomes and Cost Analysis of Fibula Free Flaps: A Retrospective Comparison of CAD/CAM versus Conventional Technique

**DOI:** 10.3390/jpm12060930

**Published:** 2022-06-07

**Authors:** Juan Pablo Rodríguez-Arias, Blanca Tapia, Marta María Pampín, Maria José Morán, Javier Gonzalez, Maria Barajas, Jose Luis Del Castillo, Carlos Navarro Cuéllar, Jose Luis Cebrian

**Affiliations:** 1Oral and Maxillofacial Surgery Department, University Hospital La Paz, Universidad Autónoma de Madrid, 28046 Madrid, Spain; mpampin@ucm.es (M.M.P.); mjose.moran@salud.madrid.org (M.J.M.); jgmartinmoro@salud.madrid.org (J.G.); mariabarajasblanco@gmail.com (M.B.); josedel.castillo@salud.madrid.org (J.L.D.C.); josel.cebrian@salud.madrid.org (J.L.C.); 2Anesthesia and Intensive Care Department, University Hospital La Paz, Universidad Autónoma de Madrid, 28046 Madrid, Spain; blanca.tapia@salud.madrid.org; 3Oral and Maxillofacial Surgery Department, Hospital General Universitario Gregorio Marañón, 28007 Madrid, Spain; cnavarrocuellar@gmail.com

**Keywords:** oral cancer, head and neck tumor, fibula free flap, virtual surgical planning, CAD/CAM, cost analysis

## Abstract

**Simple Summary:**

Current planning techniques, including computer-aided design/computer-aided manufacturing (CAD/CAM), offer ways to plan reconstructive surgery that optimize aesthetic outcomes and functional rehabilitation. In addition, the use of these techniques promotes safety and reduces the duration of operations. This study compares both the cost of CAD/CAM technology and the clinical results with those of the conventional technique in remodeling the fibula for mandibular reconstruction.

**Abstract:**

(1) Background: A decrease in operative time can not only improve patient outcomes through a reduction in the risk of developing complications but can also result in cost savings. The aim of this study is to determine whether there an intraoperative time gain can be achieved by using the preoperative virtual planning of mandibular reconstruction using a free fibula flap compared with freehand plate bending and osteotomies. (2) Methods: A retrospective comparative study was carried out in the Oral and Maxillofacial Department of La Paz University Hospital, Madrid, Spain. The study compared 18 patients in the CAD/CAM group with 19 patients in the conventional freehand group. A comparison was made between the total surgical time, the comorbidities, and the hospital stay. The resource consumption was estimated using a cost analysis. (3) Results: Although CAD/CAM was a statistically more expensive procedure in the perioperative phase, no significant differences were observed in total health care costs between the two groups. There was a non-significant trend towards an increase in complications with conventional reconstruction plates compared to patient-specific plates (PSI). (4) Conclusions: CAD/CAM technology and a 3D printed cutting guide offer a significantly shorter surgical time, which is associated with a reduction in hospital days, PACU days, and complications. The cost of CAD/CAM technology is comparable to that of the conventional freehand technique.

## 1. Introduction

The main challenge in the reconstructive surgery of the craniomaxillofacial (CMF) region is to achieve optimal function and aesthetics, despite its complex three-dimensional (3D) anatomy.

For maintaining the patient’s quality of life—in terms of both esthetics and function—the restoration of mandibular continuity after the extirpation of benign or malignant lesions is crucial.

The gold standard for the restoration of mandibular defects has become the use of the free fibula flap (FFF) because of the following: its adequate bone and pedicle length; minor donor-site morbidity; and high survival rate for both the flap and the dental implants. Since its use by Hidalgo in 1989, various techniques to reconstruct a mandible with a fibular flap have been described [1].

The problem with conventional techniques lies in trying to reconstruct a complex 3D structure by 2D imaging, or by using various intraoperative bent templates. For complex cases, it can be time-consuming, laborious, and unreliable. The selection of these techniques can, therefore, negatively influence both the functional and esthetic outcomes [2].

Current planning techniques, including computer-aided design/computer-aided manufacturing (CAD/CAM), offer ways to plan reconstructive surgery that optimize aesthetic outcomes and functional rehabilitation.

The introduction of contemporary technologies such as three-dimensional computer-assisted surgery (CAS), together with CAD/CAM, has provided reconstructive CMF surgeons with the essential tools to meet the challenges of their given surgical field. The personalized planning and modeling of cutting guides, as well as the use of pre-bent plates, allow for correct flap placement that resembles the initial form of the mandible, thus optimizing aesthetic outcomes, optimizing functional rehabilitation, and reducing the operative time [3].

This reduction in operating time could potentially lead to a better outcome for the patient while reducing overall costs. In addition, a shorter ischemia time and fewer days of hospitalization could potentially justify the higher preoperative costs related to CAS. It is also important to keep in mind that institutional costs per minute of surgery time are close to EUR 30–50 [3,4,5].

A decrease in operative time can not only improve patient outcomes by reducing the risk of developing complications, but can also result in cost savings. The probability of developing complications has been shown to increase by 14% with every 30 min increment in operating time [6].

Despite the extensive literature on the benefits of CAS as a surgical option, the cost of CAD/CAM technology is often considered an obstacle to its widespread use. Another reported disadvantage of CAS is the need for longer planning and preparation time [7].

Our study was designed to determine if the use of CAS results in a reduction in operating time and hospitalization days, and if this time gain results in a self-financing technique.

Our staff possess 20 years of experience with FFF reconstruction for mandibular or maxillary defects, performing approximately 60 free-flap procedures per year. The current study presents a single tertiary center’s 6-year experience with FFF reconstruction for mandibular defects only, and aims to determine the impact of the two surgical techniques (CAD/CAM technology versus conventional freehanded technique) on the Spanish health system. 

First, we sought to determine whether there an intraoperative time gain can be achieved by using the preoperative virtual planning of mandibular reconstruction using microvascular fibular flaps with guided surgery, cutting guides, and customized plates, as compared with freehand plate bending and osteotomies. Second, we set out to determine if this would translate into a cost reduction resulting from a lower complication rate and fewer days of hospitalization, and further, if this would translate into the self-financing of the additional expenses incurred by outsourcing the planning and cutting guides or the plate production.

## 2. Materials and Methods

A retrospective single-center cohort study was conducted at the Oral and Maxillofacial Department of Hospital La Paz, Madrid, Spain.

### 2.1. Patients

The study population comprised all adult patients (>18 years of age) who underwent microsurgical mandible reconstruction between January 2015 and December 2020. Patients were divided into two groups; Group I (CAD/CAM group) included patients who underwent mandibular reconstruction using CAS and surgical templates, and Group II (conventional group) included patients treated with freehand surgery. Patients younger than 18 years and patients who had required a previous free flap for head and neck reconstruction were excluded from this study.

A complete pre-anesthetic evaluation was performed for all included patients. All patients underwent complete laboratory tests (complete blood count, clotting times, and electrolyte levels) and routine chest radiography. All were evaluated preoperatively with submillimeter computed tomography (CT) of the head and neck area and a CT angiography of the lower extremities, with a resolution of 64 slices. All computed tomography was performed under a navigation protocol.

All surgeries were performed by the same 4 surgeons and under the same standardized anesthetic protocol.

### 2.2. CAD/CAM Group

Digital Imaging and Communications in Medicine (DICOM) files were processed at Avinent (Santpedor, Barcelona), Synthes (Basel, Switzerland), and Materialize (Leuven, Belgium) with the Pro-Plan CMF software and Web meetings.

On both the mandible and fibula, osteotomies were virtually simulated. For malignant mandibular lesions, a safety margin of 1 cm was respected.

The planning time was variable, with a maximum of 1 h, depending on the complexity of the case.

Finally, patient-specific mandibular and fibula surgical cutting templates, with STL models and PSI 2.5–2.7 mm reconstructive plates, were created within 10 business days.

### 2.3. Freehand Technique Group

The 2.5–2.7 mm reconstruction plate was manually bent intraoperatively using a pre-plating technique prior to resection of the mandibular segment [8]. Afterwards, the plate was removed, and the segments were resected. The free fibula was reshaped manually, imitating the length and curvature of the lower margin of the lower jaw. The plate was fixed with at least 3 bicortical locking screws on each side of the planned excision segment. There were no soft tissues found between the plate and the mandible of any patient, preventing flexion and resulting in a good fit of the plate on the external ridge of the mandible.

The flaps were not elevated with the use of a tourniquet for any of the groups, because our experience shows that this is not needed in free fibula flap harvesting and that there are no explicit benefits. Moreover, a tourniquet could induce microthromboses due to local compression. For all patients, the fibula flap was raised with a skin paddle, either for soft tissue support or for flap monitoring.

### 2.4. Postoperative Care

Postoperative care in the postanesthesia care unit (PACU) was the same in the 2 groups, although the length of the patients’ hospital stays varied.

### 2.5. Analysis

The parameters recorded for the test and control groups included age; sex; American Society of Anesthesiologists (ASA) score; tumor etiology; the number of bone segments; defect size; neck dissection; tracheostomy; operation time; the number of hospitalization and PACU days; complications; and the percentage of positive, close, and negative margins after surgery, defining a close margin as <4 mm and a negative margin as >5 mm.

For each patient, the following items were included in the cost analysis: the medical personnel involved (maxillofacial surgeons, anesthetists, nursing nurses, and ward nurses); the hospital stay; the stay in the PACU; surgery time; and the costs of the CAD/CAM technique (planning, cutting guides, custom plate, and screws), as well as regular reconstruction plates (plates and screws). The cost of medical staff care was expressed quantitatively by an expert clinical manager from the management service. The costs associated with anesthetic drugs were excluded.

### 2.6. Statistical Analysis

IBM SPSS 149 Statistics for Windows (version 22.0, Armonk, NY, IBM Corp) was used. The chi-square test was used to analyze the data. All parameters were analyzed using descriptive statistics and frequencies. Correlation analyses were conducted using the ×2 test or Fisher’s exact test. Multivariate analysis was performed using logistic regression and the probability ratios (OR) were calculated. A *p*-value of less than 0.05 was considered significant.

## 3. Results

### 3.1. Demographic Data

The number of patients who met the inclusion criteria was 37. For 18 of them, mandibular free-flap reconstruction using CAD/CAM technology was performed, whereas in the other 19 traditional mandibular reconstruction using a freehand bent plate for fibula flap fixation was performed. The mean age of the patients was 47 years for the CAS group and 55 years for the control group.

There were no differences between the groups in terms of demographics, medical history, or ASA, as shown in Table 1.

Differences in a patient’s diagnosis before and after reconstruction could be explained by a delay in the planned surgery. The freehand reconstruction technique tends to be used for malignant etiologies, as shown in Figure 1.

The distribution between groups according to the classification of the American Society of Anesthesiology (ASA) was homogeneous: ASA I (21% vs. 38.8%); ASA II (52.6% vs. 33.3%); ASA III (26, 3 vs. 27, 7%); and ASA IV (0% vs. 0%). There was no statistically significant presence of cardiovascular risk factors, such as diabetes mellitus or smoking. Pretreatment with RT did not differ between groups, and the rates of postoperative RT were also similar (*p* = 0.43).

A tracheotomy was performed for 70% of all patients (sixteen in the conventional group vs. eleven in the CAD/CAM group). In 54% of cases, patients had a simultaneous neck dissection (thirteen in the conventional group vs. seven in the CAD/CAM group). There was no difference between the groups in the number of patients who underwent unilateral or bilateral neck dissection (*p* = 0.341).

There were no differences between groups in the survival rate of the flaps or in the number of flap osteotomies. Furthermore, complications were not different in either group.

### 3.2. Length of Stay

The total difference in the average time gain in the OR was 102 min for the CAD/CAM group. This difference was statistically significant.

In order to determine whether other variables, such as the diagnosis, cervical dissection, or the number of fibula segments, were responsible for this difference or whether there truly was a difference between the conventional and CAD/CAM-guided surgeries, a multivariate analysis was conducted. The difference between the groups was also found to be statistically significant after multivariate analysis (Table 2).

We calculated the institutional value per minute of theatre time to be EUR 35. The typical cost of CAD/CAM reconstruction, in conjunction with the design and cutting of templates and custom-made plates, was approximately EUR 6000 per case. The value of a normal reconstructive plate was EUR 1800. By taking the value per minute into account, the money saved from the time gain is more than EUR 4200. This cost corresponded approximately to the total price of the CAD/CAM surgery. Moreover, the results showed a non-statistically significant reduction in hospitalization time by 3 days in patients with CAS, and a statistically significant reduction in time spent in the PACU by 1 day. This resulted in an extra savings of EUR 1300 (EUR 100/hospitalization day and EUR 1000/PACU day) in our hospital.

This difference corresponds to an additional cost reduction for the CAD/CAM group.

The treatment delay from the time of diagnosis to the time of OR was 12 days longer for the CAD/CAM group compared to the conventional group. No tumor progression was noted in the CAD/CAM-guided group.

### 3.3. Complications

The following results were obtained for complications following treatment, with small differences between groups (Table 3). The percentages of closed and negative margins for the conventional group were 37% and 52%, respectively, and the margins for the CAD/CAM-guided group were 18% and 72%, respectively. Two cases of positive margins were observed in each group.

The surgical reintervention rate for complications such as bleeding or infection was 16%, with no significant differences between groups. Flap survival rate, including that of salvaged flaps, was 100%. The infection rate for either the donor or acceptor site was 27%.

Three patients (8%) developed a respiratory infection, with no significant differences between groups, and the infections were treated successfully with antibiotics.

## 4. Discussion

Economic crises put pressure on public health systems to reduce costs, causing doctors and surgeons to offer the best possible treatment with the means at hand while also acting as good stewards of these means to ensure the well-being of patients. Minimizing costs can play an important role in the optimization of patient treatment.

In addition to esthetic deficits, mandibular defects can lead to severe functional deficits associated with chewing, swallowing, speaking, and breathing that negatively affect a person’s quality of life. Reconstruction is therefore an inevitable requirement.

CAD/CAM technology enables rehabilitative surgeons to perform this complex surgery with the ideal pre-surgical work-ups for neoplasm resections, surgical reproducibility for locating and targeting osteotomies, and comparatively shorter ischemia and operation times that reduce postoperative hospital stays [7].

Although the benefits of the use of CAD/CAM technology for fibular jaw reconstruction are well known, its general adoption is impeded by both the cost estimates for this technology, from EUR 4000 to EUR 5000 per patient, and a product delivery time of greater than 3 weeks [9].

Because reducing operating times can significantly affect the total cost of treatment, these savings then become available for the provision of other essential services or for the self-financing of this technique.

Other methods that can be used to reduce OR times include the following: using a two-team approach for tumor resection and flap elevation; regularizing intraoperative and postoperative care; choosing reliable flaps; accomplishing single-stage resections and reconstructions; and creating large-volume centers.

With 3D planning and guided surgery, Zweigle et al. reported a decrease in operative time by 67.4 min [7], and Nilsson et al. reported even greater savings of 84.6 min [10].

We found that the operating time was reduced by 120 min, which was a statistically significant difference even after multivariate analysis. Given the EUR 35 cost per minute in our operating room, savings of EUR 4200 (Figure 2) were realized, demonstrating that with this degree of cost savings in the OR, the CAS technique is self-financing.

Furthermore, a decrease in operative time can improve patient outcomes by reducing the risk of developing complications, since the probability of developing complications increases by 14% with every 30 min increment in operating time [6]. For CAS patients, we found 5% fewer complications.

In addition, our study showed shorter total hospital and postoperative PACU stays.

Our results showed a non-statistically significant decrease in hospitalization time by 3 days for patients with CAS, and a decrease in the time spent in the PACU by 1 day. This generates an extra savings of EUR 1300 (EUR 100/hospitalization day and EUR 1000/PACU day). CAD/CAM is not only a self-financing technology, but also very efficient.

Our results are similar to those of Tarsitano et al. and Bolzoni et al., who reported a similar total cost for the two groups. Furthermore, although the cost difference of the plates was substantial, the use of CAD/CAM technology decreased complications and improved outcomes, thereby covering this difference [5,11].

Close to 62% of mandibular reconstructions are primary reconstructions after oral squamous cell cancer (OSCC), which is the most common malignancy of the upper aerodigestive tract, with about 90–95% prevalence [12].

Surgical therapy is aimed to excise the neoplasia with a surrounding safety margin of ≥5 mm (R0-resection), corresponding to an intra-oral distance of 10 mm to the palpable tumor border.

In 2019, Goetze et al. concluded that patients with oral OSCC affecting the jaws can be safely treated via primary reconstruction with the help of CAD/CAM-guided techniques, and CAS has a 20% lower positive margin using this approach [13]. Our study shows no loss of resection margin safety with the use of this technology.

This could result from the fact that, in the operating room and with the patient attentive, even if an adequate margin is tried, it tends to be less aggressive than during virtual planning. It is easier to be more aggressive while operating on an image than on an actual body.

In addition to positive margins, prognostic factors for tumor size and increased stage, lymph node involvement (extracapsular spread), distant metastasis, and the stage of the presenting lesion at the time of diagnosis are the most important prognostic markers for oral cancer.

The mean total interval, outlined as the time from when the patient notices the initial signs and symptoms to the start of treatment, has been reported as 206 days with a range of 52 to 786 days [12]. Moreover, the patient interval is defined as the time that elapses between the moment the patient first notices the signs and/or symptoms and their first visit with a health care provider (HCP) and includes both the assessment intervals (such as those seeking help). This is the largest contributor to the full-time period, ranging from 1.6 to 5.6 months [14].

The pretreatment interval is defined as the time from diagnosis to the beginning of treatment and can be influenced by the patient, the health system, and injury factors. It is known that this time can compromise the case’s prognosis, since during this time interval the tumor has the ability to multiply and metastasize. A study conducted in 2013 in the United States (US), using the National Cancer Database, reported a mean pretreatment interval of 30 days [15].

Along the same lines, our study revealed a statistically significant prolonged time to the start of therapy, at a median of 34 days, when the CAD/CAM method was used compared with conventional method (22.0 days). We did not observe that the delay time significantly affected the soft and bony resection margins in all cancer patients.

Since the patient interval is the largest contributor to the total time interval, priority should be given to programs that aim to increase public education and awareness of the early signs and/or symptoms of OSCC.

Dell’Aversana Orabona in 2018 demonstrated that, with an in-house procedure, the time required to manufacture the customized cutting guides was much shorter when compared to the time taken by the commercial system and could potentially be reduced by 24 h. This is especially important when we treat malignant diseases. In addition, the price of the printing material is estimated at around EUR 3 [16].

Although these systems have radically changed the methods used to perform mandibular reconstructive surgery, the lack of protocols and laws that assure or control the quality of the guides that one carries out should be debated.

It is possible to discuss the degree of training that one receives for cutting plates or guides, whether or not the help of biomedical engineers should be mandatory for its realization, and the conflict between ethics and the principles of justice and charity. We believe that, without the help of engineers, we could not ensure that the best and most appropriate treatment that we have at our disposal is always offered to the patient.

Our experience has been similar to that of most studies; if we weigh the pros and cons, the main two disadvantages associated with CAD/CAM—the preoperative time for planning and the time delay between planning and receiving the hardware—do not seem to detract from the great advantages that CAS brings.

This study has two main limitations. First, the retrospective nature of the study and the subjective nature of some of our parameters does not allow for a standardized evaluation of them. Second, a relatively small sample size could have confused the results. The current study presents a single tertiary center’s 6-year experience with FFF reconstruction for mandibular defects only. A larger sample could provide a better understanding of how CAS benefits patient outcomes and quality of life.

## 5. Conclusions

For patients undergoing a mandibular reconstruction with a FFF, the use of CAD/CAM technology appears to offer significantly less time in surgery and is associated with fewer days in the PACU.

For CAS versus freehand surgery, we found a reduction in hospitalization days; PACU days; and, potentially, complications.

A reduction in operating times should be a universal goal for surgeons, hospitals, and lawmakers, not only because reducing operating times could improve patient outcomes by reducing the risk of complications, but also because reduced operating times reduce costs.

## Figures and Tables

**Figure 1 jpm-12-00930-f001:**
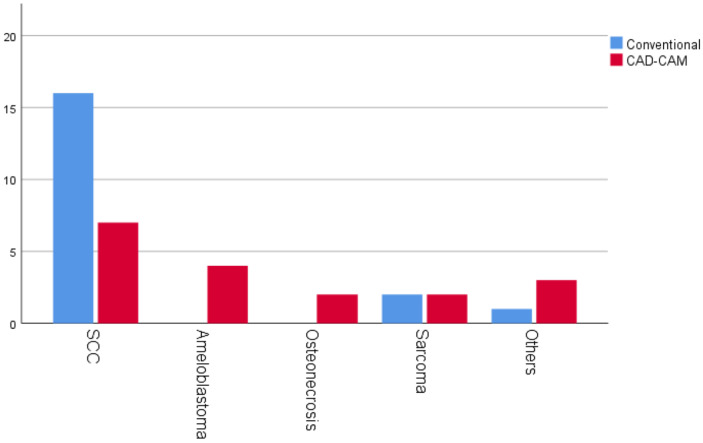
Diagnoses. SCC (squamous cell carcinoma).

**Figure 2 jpm-12-00930-f002:**
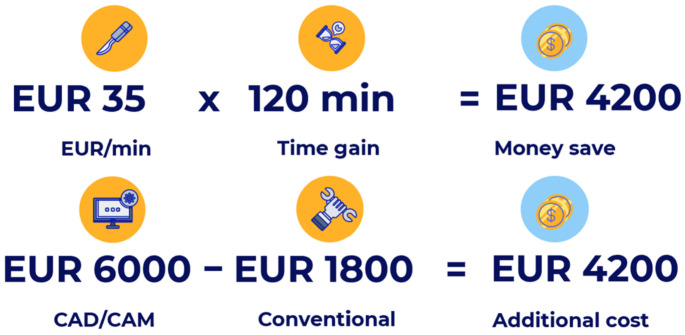
Cost savings. Cost comparison of pre-curved plates and virtual planning, taking into account the cost savings from the OR time gain.

**Table 1 jpm-12-00930-t001:** Demographic data.

		Conventional (19)	CAD/CAM (18)
Gender	Male	13	6
	Female	6	12
Mean age		55	47
ASA	I	4	7
	II	10	6
	III	5	5
Diagnosis	Squamous cell carcinoma	16	7
	Ameloblastoma	0	4
	Osteonecrosis	0	2
	Sarcoma	2	2
	Others	1	3
Fibular segments	1	7	3
	2	9	10
	3 or 3+	3	6

**Table 2 jpm-12-00930-t002:** Continuous data comparison.

	Conventional (19)	CAD/CAM (18)	*p*-Value
Days to surgery	22.5	34.6	0.039 *
Operating time (min)	623	521	0.018 *
Average hospitalization (days)	24.3	21.8	0.06
Average PACU (hours)	65.5	43	0.02 *

* These differences in time were statistically significant after uni- and multivariate analyses.

**Table 3 jpm-12-00930-t003:** Analyses of complications.

	Conventional (19)	CAD/CAM (18)	*p*-Value
Reinterventions	2	4	- *
Infections	4	6	- *
Pulmonary Embolism (PE)	1	0	- *

* We were unable to obtain a *p*-value due to the low number of cases in both groups.

## Data Availability

The data presented in this study are available on request from the corresponding author.

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
