# Peer review of "Clinical Outcomes and Cost Analysis of Fibula Free Flaps: A Retrospective Comparison of CAD/CAM versus Conventional Technique"

_jpm, 2022, doi:10.3390/jpm12060930_

Round 1

Reviewer 1 Report

Overall an excellent paper. Please  find below few clarifications and suggestions:

  1. references are required for the statement in Introduction "The personalized planning and modeling of cutting guides, as well as the use of pre-bent plates, allow correct flap placement, resembling the initial form of the mandible, optimizing aesthetic outcomes and functional rehabilitation, and reducing operative time" Are there existing articles that have previously documented this.
  2.  Since there was significant difference in the distribution of malignancies between the two groups. Did the authors take into account the confounding factor of time taken for neck dissection?
  3. As mentioned in the discussion, "product delivery times greater than 3 weeks", was tumour progression noted in the malignancies operated by CAD/CAM technique

Author Response

Thank you for your review time.

  1. References are required for the statement in Introduction "The personalized planning and modeling of cutting guides, as well as the use of pre-bent plates, allow correct flap placement, resembling the initial form of the mandible, optimizing aesthetic outcomes and functional rehabilitation, and reducing operative time" Are there existing articles that have previously documented this.

Reference added.

  1. Since there was significant difference in the distribution of malignancies between the two groups. Did the authors take into account the confounding factor of time taken for neck dissection?

As we say in line, only 54% of patients had a simultaneus neck dissection, with 6 patients more in the CAD-CAM-guided group, but we didn´t find statistically difference (p=0,341). In fact, we underwent a multivariate analyses. (Table2). I have tried to point this out in line 194: "This difference between the groups was statistically significant after multivariate analysis, including factors such as having a simultaneous neck dissection or the number of fibular segments. "

  1. As mentioned in the discussion, "product delivery times greater than 3 weeks", was tumour progression noted in the malignancies operated by CAD/CAM technique

As we treat in a public hospital, where the availability of the operating room is not so much, we only found a difference in the pre-treatment interval of 12 days, without finding tumor progression. Added in line 210

I remain at your disposal, open to any other suggestion or need for change.

Reviewer 2 Report

The scientific work is clearly written and fulfills the purpose of the journal. It addresses a topical issue for the medical community which is to ensure personalized treatment for each patient in compliance with sustainable healthcare costs.

The hypothesis to be validated is that the use of CAD-CAM technology, compared to the conventional technique, considerably reduces the operating times and the length of hospital stay of patients undergoing mandibular reconstruction surgery with FFF, making the two procedures equivalent in terms of overall costs.

The analysis of the literature on the various points of the discussion is sufficiently complete and updated. Both the introduction and the discussion address the proposed theme in a relevant way.

However, there are some weaknesses in the study that could make it inappropriate to test the hypothesis.

Major revisions:

1) The sample is inhomogeneous concerning the initial diagnosis and this adversely affects the study design. Squamous cell carcinoma was diagnosed in 16/19 patients in the conventional group and only in 7/18 in the CAD-CAM group. This data implies that in 13/19 patients of the conventional group a lateral cervical lymph node dissection was performed and in 16/19 a tracheostomy, with significant differences compared to the CAD-CAM group in which neck dissection was performed in 7/18 and a tracheostomy in 11/18. In the conventional group, six laterocervical dissections and five more tracheotomies were performed, than in the CAD-CAM group. Both of these additional surgical procedures objectively require time to be performed by the surgical team and it is not clear whether the Authors took into account these important differences in calculating the average time gain of 102 min of the CAD-CAM technique, compared to the conventional.

2) As regards the analysis of the costs of medical personnel, hospital stays, PACU, and surgery time, these were quantified by the Authors thanks to the support of the hospital's management service (35 euros for a minute of theater time, 100 euros for each hospitalization day and 1000 euros for each PACU day). However, these quantitative values are too subjective and difficult to compare on a general scale as they are subject to too much variability to constitute an objective parameter for comparison and would require standardization.

3) The sample size of the retrospective analysis is also too small (37 patients) and does not provide significant results on the clinical outcomes of the two techniques in comparison. The analysis of the data relating to the complications of the two groups is lacking. Unlike what has been stated, the number of complications (reoperations and infections) prevails in percentage terms in the CAD-CAM sample, albeit in the absence of statistical significance. Furthermore, data on postoperative functional results are not analyzed and the patient follow-up period is not indicated.

4) Finally, the negative criticism of the use of cheaper in-house procedures for the creation of customized cutting guides is too personal and subjective and is not supported by scientific evidence.

Minor revisions:

Lines 31-32 “the CAD-CAM…included 19 patients” to be deleted because already mentioned in the methods.

Line 37 “that allow better clinical and aesthetic results” to be canceled because it is neither demonstrated nor analyzed in the study.

Lines 137-141 are to be canceled because it is not relevant to the study.

Line 149 “and postoperative functional results” to be deleted because not reported in the results.

Line 2017 “with no significant differences between groups” to be deleted as inaccurate.

Lines 279-280 “but our study shows no loss of safety of the resection margins” to be cleared or better specified in materials and methods and results.

Lines 284-287 are to be canceled because it is not relevant to the study

Line 335 “and also fewer complications” to be deleted as inaccurate.

Line 338 “and fewer complications” to be deleted as inaccurate.

Author Response

Thank you for your time.

I remain at your disposal, open to any other suggestion or need for change.

Major revisions:

  • The sample is inhomogeneous concerning the initial diagnosis and this adversely affects the study design. Squamous cell carcinoma was diagnosed in 16/19 patients in the conventional group and only in 7/18 in the CAD-CAM group. This data implies that in 13/19 patients of the conventional group a lateral cervical lymph node dissection was performed and in 16/19 a tracheostomy, with significant differences compared to the CAD-CAM group in which neck dissection was performed in 7/18 and a tracheostomy in 11/18. In the conventional group, six laterocervical dissections and five more tracheotomies were performed, than in the CAD-CAM group. Both of these additional surgical procedures objectively require time to be performed by the surgical team and it is not clear whether the Authors took into account these important differences in calculating the average time gain of 102 min of the CAD-CAM technique, compared to the conventional.

It is true that, as this was a retrospective study and, as an inherent bias, it was probably chosen to plan CAD-CAM surgery in patients with benign neoplasms as a precaution against the possible delay in providing the cutting guides by the commercial companies.

However, as we say in the line, only 54% of patients underwent simultaneous neck dissection, with 6 more patients in the CAD-CAM-guided group, but we found no statistical difference (p=0.341). In fact, we performed a multivariate analysis (Table 2). I have tried to point this out in line 194: "This difference between the groups was statistically significant after multivariate analysis, including factors such as having a simultaneous neck dissection or the number of fibula segments. "

  • As regards the analysis of the costs of medical personnel, hospital stays, PACU, and surgery time, these were quantified by the Authors thanks to the support of the hospital's management service (35 euros for a minute of theater time, 100 euros for each hospitalization day and 1000 euros for each PACU day). However, these quantitative values are too subjective and difficult to compare on a general scale as they are subject to too much variability to constitute an objective parameter for comparison and would require standardization.

Although it is not a detailed analysis that accurately breaks down the costs, we believe that it is a sufficiently objective and valid analysis to make such a comparison. It is true that it can be done more accurately, and it is our intention, at a later stage, to complete the study with more patients, and with a more accurate study. However, our data are within the mean published in the literature and are expressed in the same units and values.  You will find that the costs of our form are provided in articles such as Tarsitano or Zweifel.  Line 71: "It is also important to keep in mind that institutional costs per minute of surgery time are close to € 30-50. [3-5]"

  1. Tarsitano, A. Is a computer-assisted design and computer-assisted manufacturing method for mandibular reconstruction economically viable?, Journal of Cranio-Maxillo-Facial Surgery. 2016; http://dx.doi.org/10.1016/j.jcms.2016.04.003 
  2. Zweifel, D. F.; Simon, C. Are Virtual Planning and Guided Surgery for Head and Neck Reconstruction Economically Viable? Journal of Oral and Maxillofacial Surgery. 2014; 73(1), 170 175. doi:10.1016/j.joms.2014.07.038

  • The sample size of the retrospective analysis is also too small (37 patients) and does not provide significant results on the clinical outcomes of the two techniques in comparison. The analysis of the data relating to the complications of the two groups is lacking. Unlike what has been stated, the number of complications (reoperations and infections) prevails in percentage terms in the CAD-CAM sample, albeit in the absence of statistical significance. Furthermore, data on postoperative functional results are not analyzed and the patient follow-up period is not indicated.

I have removed from the text everything related to functional results, which we finally decided not to include in the paper.

It is true that we provide a small number of patients and that, since these patients suffer few complications, no conclusions can be drawn in this regard. It is discussed in the limitations of the study.

I have removed anything that could lead to error, as in the conclusions, in this sense.

4) Finally, the negative criticism of the use of cheaper in-house procedures for the creation of customized cutting guides is too personal and subjective and is not supported by scientific evidence.

I am sorry if this is understood as criticism. It is simply a call for reflection on a topic that, in fact, is already being debated, such as whether it is ethical to plan these technologies without the help of an engineer and whether there is a regulation that regulates or supervises this.

In fact, there are softwares that are not approved. For example, to quote you an article: “There are limitations to the in-house CAD/CAM workflow. Although the technology supports the design and fabrication of sterilizable models and surgical guides because of current regulatory standards in the United States, the use of open source software and non-FDA cleared 3D printers is limited to the use of modified anatomic models as nonsterile surgical references at the present time.” (Moe, J., Foss, J., Herster, R., Powell, C., Helman, J., Ward, B. B., & VanKoevering, K. (2020). An In-House CAD/CAM Workflow for Maxillofacial Free-Flap Reconstruction is Associated with A Low Cost and High Accuracy. Journal of Oral and Maxillofacial Surgery. doi:10.1016/j.joms.2020.07.216 )

Minor revisions:

Lines 31-32 “the CAD-CAM…included 19 patients” to be deleted because already mentioned in the methods.

Delated

Line 37 “that allow better clinical and aesthetic results” to be canceled because it is neither demonstrated nor analyzed in the study.

Delated

Lines 137-141 are to be canceled because it is not relevant to the study.

These lines are pointed out because in many centers the flaps are removed with ischemia. We point this out to describe the technique we chose and why we did not analyze the ischemia time. It is known that there is a theoretical limit of ischemia of 5h, but we do not write more about it in the article precisely because we do not use the ischemia technique when harvesting the flap.

Line 149 “and postoperative functional results” to be deleted because not reported in the results.

Delated.

Line 2017 “with no significant differences between groups” to be deleted as inaccurate.

Delated

Lines 279-280 “but our study shows no loss of safety of the resection margins” to be cleared or better specified in materials and methods and results.

Corrected. Lines 148 and 215

Lines 284-287 are to be canceled because it is not relevant to the study

We believe it is important as a link between the prognostic factors we discussed, such as positive margins or pre-treatment time intervals.

Line 335 “and also fewer complications” to be deleted as inaccurate.

Delated

Line 338 “and fewer complications” to be deleted as inaccurate.

Corrected

Round 2

Reviewer 2 Report

Thank you very much and I am satisfied with your answers to my questions. I only ask you to expand and clarify in your scientific work what is expressed in line 194 regarding the multivariate analysis.

Author Response

Good afternoon, I have tried to explain it better.
Write to me if it is not enough.

Kind regards

This manuscript is a resubmission of an earlier submission. The following is a list of the peer review reports and author responses from that submission.